# Effects of Disruption of Five *FUM* Genes on Fumonisin Biosynthesis and Pathogenicity in *Fusarium proliferatum*

**DOI:** 10.3390/toxins11060327

**Published:** 2019-06-07

**Authors:** Lei Sun, Xu Chen, Jian Gao, Yuan Zhao, Lianmeng Liu, Yuxuan Hou, Ling Wang, Shiwen Huang

**Affiliations:** 1State Key Laboratory of Rice Biology, China National Rice Research Institute, Hangzhou 310006, China; leisun248@126.com (L.S.); chenxu0227@126.com (X.C.); JIANGAO221214@126.com (J.G.); zmy260650584@163.com (Y.Z.); liulianmeng@caas.cn (L.L.); houyuxuan@caas.cn (Y.H.); 2College of Agronomy, Guangxi University, Nanning 530003, China

**Keywords:** *Fusarium proliferatum*, *FUM*, fumonisin, pathogenicity

## Abstract

The mycotoxin fumonisin is known to be harmful to humans and animals, and thus it is desirable to reduce fumonisin content in crop products. We explored the functions of several genes that function in fumonisin biosynthesis (*FUM1*, *FUM6*, *FUM8*, *FUM19*, and *FUM21*) in *Fusarium proliferatum* and found that deletion of *FUM1*, *FUM6*, *FUM8*, or *FUM21* results in a severe reduction in fumonisin biosynthesis, while loss of *FUM19* does not. In addition, fumonisin-deficient strains display significantly decreased pathogenicity. Co-cultivation of the Δ*FUM1*, Δ*FUM6*, Δ*FUM8*, and Δ*FUM19* mutants restores fumonisin synthesis. However, co-cultivation was unable to restore fumonisin synthesis in the Δ*FUM21* strain. The relative expression levels of three key *FUM* genes (*FUM1*, *FUM6*, and *FUM8*) differed significantly in each mutant strain; notably, the expression levels of these three genes were significantly down-regulated in the Δ*FUM21* strain. Taken together, our results demonstrate that *FUM1*, *FUM6*, *FUM8*, and *FUM21* are essential for fumonisin synthesis, and *FUM19* is non-essential. Partial mutants lost the ability to synthesize fumonisin, the co-culture of the mutants was able to restore fumonisin biosynthesis. While the pathogenicity of *F. proliferatum* is affected by many factors, inhibition of the synthesis of the mycotoxin fumonisin will weaken the pathogenicity of rice spikelet rot disease (RSRD).

## 1. Introduction

The filamentous ascomycete *Fusarium proliferatum* (teleomorph *Gibberella intermedia*) infects commercially important plants such as rice, maize, wheat, barley, banana, and mango, resulting in losses in production quantity and crop quality [1,2,3,4,5,6,7]. For example, in rice production, the annual average planting area of japonica rice varieties in China is approximately 7.5–8.4 million hm^2^, and the annual incidence area of RSRD is 0.8–1 million hm^2^. The seed setting rate of infected rice is reduced by 8–10%, and the 1000-grain weight is reduced by 0.6–1.0 g. The yield is generally reduced by 5–10%, and the yield of severely infected rice plants is reduced by more than 30% [8]. 

*F. proliferatum* is able to produce a variety of mycotoxins, including fumonisins, moniliformin, beauvericin, fusaric acid, fusaproliferin, and bikaverin [9,10,11,12,13]. Among them, fumonisins are some of the most harmful fungal toxins and exert hepatotoxic, nephrotoxic, hepatocarcinogenic, and cytotoxic effects in mammals, including humans [14,15]. B-type fumonisins are the most toxic. Fumonisin B_1_ (FB_1_) accounts for almost 70% of the total *F. proliferatum* fumonisin content, while FB_2_ and FB_3_ comprise approximately 10–20% [16]. Therefore, in our study, we chose to use FB_1_ as a representative of total fumonisin content.

A cluster of 17 fumonisin biosynthetic genes (*FUM*) have been identified and characterized in *Fusarium verticillioides*; these co-expressed genes include a gene encoding polyketide synthase (PKS), two genes encoding fatty acid synthases, nine genes encoding cytochrome P450 monooxygenases, dehydrogenases, transporter proteins, an aminotransferase, and a dioxygenase [17,18,19]. FUM1p was identified as the key enzyme of fumonisin biosynthesis, and the cluster-specific Zn(II)2Cys6 transcription factor (TF) Fum21p was found along with a putative TF-binding motif (CGGMTA) to participate in transcriptional activation of *FUM1* [17,20,21,22,23]. The *FUM19* gene is located approximately 35 kb downstream from the *FUM1* gene and encodes an ABC (ATP-binding cassette) transporter involved in extracellular export of fumonisins. *FUM8* encodes an aminotransferase required to create mature, biologically active FB_1_ [17,24,25,26]. The enzyme encoded by *FUM6* catalyzes the hydroxylation of C-14 and C-15 of fumonisin to generate early intermediates of fumonisin biosynthesis [17,25,27]. *FUM21* is a regulator of fumonisin synthesis. *FUM1*, *FUM6*, and *FUM8* are key genes in the synthesis of fumonisin precursors and *FUM19* functions in fumonisin transport. In this study, we chose to investigate these five genes to broadly understand the function of fumonisin synthesis genes and observe the effects of disruption of these genes on various aspects of fumonisin synthesis.

The *FUM* cluster has been identified in several fumonisin producers [20,22,28]. A comparative genomic approach was used in *F. proliferatum* to identify the *FUM* cluster, revealing the same order and orientation of genes as described for *F. verticillioides* and *F. oxysporum* [17,28,29]. The low level of protein sequence identity (77–89%) of *FUM* genes and the different genomic locations of the cluster in *F. proliferatum* and *F. verticillioides* indicate that each species may have acquired the cluster independently [29]. 

In recent years, the Yangtze River region of China has experienced a high incidence of rice spikelet rot disease (RSRD) [7]. RSRD causes rice grains to rot and discolor, and results in grain deformations and reductions in grain yield [7,30]. The disease is characterized by rust red or yellowish-brown oval spots on the glumes during the early growth stage, which change to brown, yellowish brown, or dark brown lesions in the later stages [7,30]. In our previous study, we assessed the most effective stage for injection of *F. proliferatum* into rice from the growing stage of pollen cell meiosis to the maturing stage. We found that the initial infection occurred during the pollen cell maturity stage, and the primary invasion sites were the anthers of rice [13,31]. It was noted that the pathogen mainly damaged the pollen cells, and with the exception of the filaments, proceeded to colonize the pistils and endosperm [31]. We investigate the function of *FUM* genes (*FUM1*, *FUM6*, *FUM8*, *FUM19*, and *FUM21*) in fumonisin biosynthesis in order to support efforts to reduce fumonisin contamination that may harm humans and animals and to protect plants from infection by fungal pathogens.

Most of the previous studies of fumonisin were performed in *F. verticillioides* [21,32,33] and are related to the function of fumonisin synthesis genes [27,34], methods of detection of fumonisin [35,36,37], and the regulation of fumonisin synthesis factors [38,39,40,41]. However, there are few studies of fumonisin production in *F. proliferatum*. The objectives of this study were to compare the functions of *FUM* genes (*FUM1*, *FUM6*, *FUM8*, *FUM19*, and *FUM21*) in *F. proliferatum* with those in *F. verticillioides*, evaluate changes in fumonisin synthesis and expression levels of key *FUM* genes in *FUM* mutant strains, and assess whether fumonisin production by *F. proliferatum* is necessary for development of RSRD symptoms. 

## 2. Results

### 2.1. Mutant Acquisition 

In order to better understand the function of *FUM1*, *FUM6*, *FUM8*, *FUM19*, and *FUM21* in fumonisin synthesis, we utilized a gene knockout approach. Deletion mutants of *FUM1*, *FUM6*, *FUM8*, *FUM19*, and *FUM21* were successfully generated through the protoplast PEG-mediated method using binary replacement vectors containing the hygromycin resistance gene. We successfully obtained five *FUM* gene disruption mutants. Figure 1 shows the construction of the DNA fragment used for knockout of *FUM21* (Figure 1A) and Southern blot data for the resulting *FUM21* deletion mutants (Figure 1D). A total of seven transformants were obtained by hygromycin resistance screening and were detected by PCR with *HygR*- and *FUM21*-YZ-specific primers. Strains 1–5 produced a 1357-bp *HygR* amplification product but not a 714-bp *FUM21* amplification product (Figure 1B), and were thus considered positive transformants. After sub-culturing for five generations, we used an additional pair of primers (*FUM21*-YZS) to PCR across part of the hygromycin resistance gene and the homologous recombination sequence upstream of the gene to further verify loss of the gene of interest. Using this approach, we found that the five mutants stably inherited their respective gene deletions (Figure 1C). We chose three strains for Southern blot analysis, which confirmed that the three *FUM21* mutants contained a single copy of the vector and were thus single-copy knockout mutants (Figure 1D).

### 2.2. Phenotypes of F. proliferatum FUM Mutants 

In order to assess whether loss of *FUM1*, *FUM6*, *FUM8*, *FUM19*, and *FUM21* affects *F. proliferatum* growth, we performed several growth experiments. When grown on PDA medium for three days, the *FUM* mutant strains displayed no significant differences in colony morphology when compared to the wild-type strain (Figure 2A–L). However, the Δ*FUM1* and Δ*FUM19* strains displayed 14.8% and 14.2% slower radial growth, respectively (Figure 2A–L and Figure 3A). After incubation for 16 h, the mutants displayed no significant changes in morphology of the mycelium at the colony edge when compared to the wild-type strain (Figure 2M–R). Δ*FUM1* and Δ*FUM19* displayed increased conidia production (approximately two to three times that of the wild-type) when grown in YEPD liquid medium for 24 h (Figure 3B). 

### 2.3. Effects of FUM Gene Disruption on Fumonisin Synthesis in F. proliferatum 

As *FUM1*, *FUM6*, *FUM8*, *FUM19*, and *FUM21* all influence fumonisin synthesis, we next assessed the ability of each deletion strain to synthesize fumonisin. It is important to note that the *FUM* mutant exhibit significantly reduced fumonisin synthesis when compared to the wild-type (Figure 4). The Δ*FUM1*, Δ*FUM6*, Δ*FUM8*, and Δ*FUM21* strains lost almost all of their ability to synthesize fumonisin when grown on RG medium. The Δ*FUM19* strain was able to synthesize 95.43 mg/kg of fumonisin, which represented only a 14.7% reduction in fumonisin synthesis when compared to the wild-type (111.83 mg/kg). 

### 2.4. Effects of FUM Gene Disruption on F. proliferatum Virulence

In other crops, fumonisins promote disease occurrence. Several of our *F. proliferatum FUM* mutants were unable to synthesize fumonisin. Therefore, we inoculated rice with our mutants to test whether fumonisin synthesis ability affects RSRD occurrence. We observed that inoculation with Δ*FUM19* caused significant deterioration of the rice spikelet after 14 days, while the other four mutants failed to symptoms (Figure 5). Analysis of the disease index 14 days after inoculation revealed that the pathogenicity of each mutant was significantly lower than that of the wild-type strain (Figure 6). In particular, the disease indexes of four mutants (Δ*FUM1*, Δ*FUM6*, Δ*FUM8*, and Δ*FUM21*) decreased 69.3%, 78.3%, 75.4%, and 78.6% respectively. 

### 2.5. Fumonisin Synthesis in Co-cultured F. proliferatum FUM Mutants

In order to better understand how *FUM* genes interact in fumonisin synthesis, we co-cultured the mutants and assessed fumonisin levels. No fumonisin synthesis was observed when Δ*FUM21* was co-cultured with each of the other three fumonisin-deficient mutants (Δ*FUM1*, Δ*FUM6*, Δ*FUM8*) in a 1:1 ratio. However, Δ*FUM21* co-cultured with Δ*FUM19* (1:1 ratio) was able to produce fumonisin; this co-culture produced 72.3% of the fumonisin produced by Δ*FUM19* cultured alone. In contrast, three other co-cultures (Δ*FUM1* + Δ*FUM19*, Δ*FUM6* + Δ*FUM19*, and Δ*FUM8* + Δ*FUM19*; all in 1:1 ratios) produced 133.0%, 177.4%, and 155.4% of the fumonisin produced by Δ*FUM19* cultured alone, greater than that observed for the Δ*FUM19* + Δ*FUM21* co-culture. In Δ*FUM1* + Δ*FUM6*, Δ*FUM1 +* Δ*FUM8*, and *FUM6* + Δ*FUM8* co-cultures, fumonisin synthesis capacities were recovered to 42.1%, 9.6%, and 59.6%, respectively, of that of the wild-type control strain (Figure 7). 

### 2.6. F. proliferatum FUM Gene Expression in Synthetic Media

From our results, we can conclude that *FUM1*, *FUM6*, and *FUM8* are important for fumonisin synthesis, and that these genes’ effects are somewhat interdependent. Therefore, we chose to quantitatively assess the expression of these three genes to test if the observed changes in fumonisin synthesis in these mutants were due to regulatory changes in gene expression. We evaluated expression of *FUM1*, *FUM6*, and *FUM8* in the wild-type and mutant strains using RT-PCR. The wild-type strains were sampled at different time points after inoculation, and transcript levels were evaluated. We found that transcript levels were significantly different across the sampled time points, and the three genes displayed their maximum expression at 48 hours (Figure 8). Therefore, we chose 48 h as our sampling time point for subsequent experiments.

*FUM* gene expression in the mutants was compared to that of the wild-type strain (Figure 9). In the Δ*FUM1* strain, the relative expression of *FUM6* and *FUM8* increased significantly by 162% and 118% of the wild-type level, respectively. From this result, we conclude that expression of *FUM6* and *FUM8* is independent of the *FUM1* gene, and deletion of *FUM1* promotes the expression of *FUM6* and *FUM8*. Deletion of *FUM6* significantly reduced expression of *FUM1* and *FUM8* by 50% and 56%, respectively. Similarly, the Δ*FUM8* strain displayed significantly reduced expression of *FUM1* and *FUM6* (by 30% and 56%, respectively). Together, these results indicate that deletion of *FUM6* or *FUM8* inhibits the expression of *FUM1*, and deletion of *FUM6* or *FUM8* inhibits expression of the other. In the Δ*FUM19* strain, the relative expression of *FUM1* increased by 90%, but *FUM6* and *FUM8* displayed no significant differences in expression, indicating that deletion of *FUM19* has no effect on *FUM6* and *FUM8* expression but can promote *FUM1* expression. In the Δ*FUM21* strain, the relative expression levels of *FUM1*, *FUM6*, and *FUM8* were significantly decreased by 20%, 36%, and 46%, respectively, indicating that strains without functional *FUM21* lacked transcripts from three biosynthetic genes (*FUM1*, *FUM6*, and *FUM8*).

## 3. Discussion

In this work, we utilized a gene knockout approach to investigate the functions and interactions of *FUM* genes in *F. proliferatum*. Deletion of the evaluated *FUM* genes has no effect on the morphology of the mycelium at the edge of the colony, but deletion of *FUM1* and *FUM19* influence growth and conidiation, likely via changes in metabolism or certain growth-related genes. Therefore, given the increase in sporulation observed for the *FUM* mutants, we speculate that *FUM1* and *FUM19* may negatively regulate sporulation. 

*FUM21* was discovered by Brown et al., who determined its location adjacent to *FUM1* and found that it encoded a zinc cluster protein that regulates transcription [22]. The authors demonstrated that strains without functional *FUM21* lacked transcripts from two biosynthetic genes (*FUM1* and *FUM8*) and failed to produce fumonisin. Similar results were found in another *Fusarium* toxin study that identified *TRI6* and *TRI10* as transcription factors in the trichothecenes biosynthetic pathway; the mutants exhibited greatly reduced pathogenicity and toxin production as well as reduced transcript levels for enzymes involved in the synthesis of the immediate molecular precursor of trichothecenes [42]. In our study, we observed that the Δ*FUM21* mutant was unable to synthesize fumonisin on solid RG medium and displayed decreased expression of *FUM1*, *FUM6*, and *FUM8* in liquid GAPL medium. When Δ*FUM21* was co-cultured with Δ*FUM19* (1:1 ratio), fumonisin synthesis was significantly reduced compared with that of Δ*FUM19* alone. However, when the other mutants were co-cultured with Δ*FUM19*, fumonisin synthesis was significantly increased. Therefore, fumonisin synthesis in the Δ*FUM21* strain cannot be restored by co-culture with Δ*FUM19*. In summary, when the Δ*FUM21* mutant was co-cultured with the other four mutants, we observed no significant changes in fumonisin synthesis, demonstrating that *FUM21* is necessary for, but not directly involved in, fumonisin synthesis. Deletion of *FUM21* affects the expression of genes directly involved in fumonisin synthesis. Therefore, we speculate that *FUM21* may play a role in regulating genes directly involved in fumonisin synthesis.

*FUM1*, which is required for fumonisin synthesis, encodes a polyketide synthase that is predicted to catalyze the initial step in fumonisin biosynthesis [25], the creation of a linear 18-carbon polyketide that forms C-3–C-20 of the fumonisin backbone [17,20]. In another study, a Δ*FUM1* mutant was unable to produce any fumonisin or new analogs, suggesting that the intrinsic interactions between the intact PKS and downstream enzymes in the biosynthetic pathway may play a role in the control of reduced polyketides in the fungus [23]. In our study, we found that the Δ*FUM1* mutant could not synthesize fumonisin, which is consistent with previous studies [20]. When the Δ*FUM1* mutant was co-cultured with the Δ*FUM6* and Δ*FUM8* mutants, both combinations restored fumonisin synthesis. This result differed slightly from those of previous studies. Others have observed fumonisin production by Δ*FUM1* or Δ*FUM8* mutants when they were co-cultured with the Δ*FUM6* mutant, but no fumonisins have been produced when Δ*FUM1* and Δ*FUM8* mutants were co-cultured with each other [25]. However, in our study, the amount of fumonisin synthesized by Δ*FUM1* + Δ*FUM8* was approximately 20% that of Δ*FUM1* + Δ*FUM6*. This result is consistent with the function of the *FUM* genes. *FUM8* is the only fumonisin synthesis gene that functions as a transaminase, and *FUM12* and *FUM15* are able to partially compensate for the loss of *FUM6* function, because FUM6p, FUM12p, and FUM15p putatively function as cytochrome P450 monooxygenases [17,32]. Therefore, the Δ*FUM1* + Δ*FUM8* co-culture synthesizes less fumonisin than Δ*FUM1* + Δ*FUM6*. 

We speculate that the restoration of fumonisin synthesis in co-cultures of *FUM* mutants is due to the exchange of various enzymes or intermediates required for fumonisin synthesis between the different mutants, which compensates for the loss of function in each mutant. However, understanding the mechanisms underlying this possible exchange between mutants will require further investigation. Previous studies identified the early fumonisin biosynthetic intermediates from fungal strains with an inactivated *FUM6* gene were purified using mass spectrometry and NMR spectroscopy [27]. In precursor feeding experiments, the fed intermediate was transformed into fumonisins by a *F. verticillioides* strain with an inactive fumonisin polyketide synthase gene, which supports our hypothesis [27]. When the Δ*FUM1* mutant was cultured in liquid GAPL medium, expression levels of *FUM6* and *FUM8* were up-regulated. Based on the above findings, we conclude that *FUM21* exerts a positive regulatory effect on *FUM1*, *FUM6*, and *FUM8*, such that when *FUM1* is deleted, the regulation of *FUM6* and *FUM8* by FUM21p is enhanced. Moreover, GAPL medium is able to induce fumonisin synthesis and thereby induce expression of the *FUM21* gene, resulting in up-regulation of *FUM6* and *FUM8*.

Fumonisins were not produced in single-strain cultures of Δ*FUM6* or Δ*FUM8*. FUM6p displays the highest level of amino acid sequence similar to a small and unusual class of enzymes that consist of cytochrome P450 monooxygenases fused to an NADPH-dependent P450 reductase [18,27]. Such enzymes are typically hydroxylases that utilize O_2_ as a source of oxygen atoms in the formation of hydroxyl groups, and thus FUM6p may catalyze hydroxylation of fumonisin C-14 and C-15 [17,27]. FUM8p is predicted to be a member of the class-II α-aminotransferases, a group of pyridoxal phosphate-dependent enzymes that catalyze the condensation of amino acids and acyl–CoAs [17,24,27]. Deletion of *FUM6* or *FUM8* inhibits the normal pathway of fumonisin biosynthesis in *F. proliferatum*. Other studies have obtained similar results, indicating that *FUM6* and *FUM8* are responsible for reactions that occur early in fumonisin biosynthesis, prior to C-14 and C-15 hydroxylation [18]. This hypothesis is consistent with the predicted functions of the encoded enzymes. When the Δ*FUM6* or Δ*FUM8* mutant was cultured in liquid GAPL medium, the expression levels of *FUM6* or *FUM8* and *FUM1* were down-regulated. This may be due to feedback inhibition, where excess intermediates inhibit the expression of upstream synthetic genes.

*FUM19* encodes an ABC transporter involved in extracellular export of fumonisins, and is not essential for fumonisin production [17]. Therefore, deletion of *FUM19* displayed little effect on the synthesis of fumonisin, which is consistent with findings of previous studies [17]. The efflux pump activity of some ABC transporters can transport compounds from inside cells to the surrounding environment, thereby reducing cellular concentrations of toxic molecules and conferring protection from them [17,26]. When the Δ*FUM19* mutant was cultured in liquid GAPL medium, the expression of *FUM6* and *FUM8* exhibited no significant changes, but expression of *FUM1* was up-regulated by 90%. A possible explanation for this result is that *FUM19* exerts a negative regulatory effect on *FUM1*, such that when *FUM19* is deleted, *FUM1* expression cannot be suppressed, resulting in significant up-regulation. However, *FUM6* and *FUM8* may not function in a signaling pathway with *FUM1* and *FUM19*, or the expression levels of *FUM6* and *FUM8* genes have reached their maximum, such that no changes could be detected. 

Mycotoxins confer a biological advantage, such as antibiosis, against competing microbes or pathogenicity of the host plant [33]. In the current study, we anticipated that fumonisin is required for virulence and host-tissue colonization. In *F. verticillioides*, fumonisin production is required for development of foliar disease symptoms on maize seedlings [33]. The *Fusarium graminearum* trichothecene precursor synthase gene *TRI5* is the first and most well-studied pathogenicity factor, and is a key enzyme in the first step of the synthesis of deoxynivalenol (DON), another important secondary metabolite. The Δ*TRI5* mutant displays significantly reduced pathogenicity in plants. These results confirm that DON production plays a significant role in the spread of fusarium bead blight within a spike, and that DON production is not necessary for initial infection by the fungus [43]. In another study, the authors found that trichothecene biosynthesis is specifically induced in infection structures, but is not a prerequisite for their development and the initial penetration of wheat tissues [44]. In other studies of fumonisin, it was found that the pathogenicities of wild strains on corn were not necessarily related to their ability to synthesize fumonisin, but the maize varieties not sensitive to FB_1_ show resistance to the infected strain [45,46,47]. FB_1_ in Fumonisins may be due to its structural similarity to sphingosine, which affects the biosynthesis of sphingolipids, eventually leading to programmed cell death in plants [48,49,50]. In our study, the four strains that displayed this considerable decline in pathogenicity were the same strains that lost their ability to synthesize fumonisin. Therefore, we speculate that loss of fumonisin synthesis is related to the decline in pathogenicity of these four mutants. After comprehensive analysis, we believe that the pathogenicity of *F. proliferatum* is affected by many factors, but inhibition of the synthesis of some secondary metabolites will weaken the pathogenicity. 

In our study, we found that *FUM1*, *FUM6*, *FUM8*, and *FUM21* are essential for fumonisin synthesis in *F. proliferatum*, while *FUM19* is non-essential. Loss of fumonisin synthesis is associated with a decrease in pathogenicity. Partial mutant co-culture can restore fumonisin synthesis. Whether this restoration is due to an exchange of fumonisin intermediates, and how these intermediates are utilized by each strain, requires further research. Our comprehensive analysis suggests that the functions of these five *FUM* genes in *F. proliferatum* are consistent with those observed in *F. verticillioides*. In future research, we will utilize overexpression of FUM proteins and assessment of additional *FUM* genes to further describe *FUM* gene functions and the fumonisin biosynthetic pathway. This research will support efforts to reduce fumonisin contamination in food production.

## 4. Materials and Methods

### 4.1. Fungal Strains, Media, Conidiation, and Growth Conditions

The FP9 strain of *F. proliferatum* was isolated from samples of rice spikelet rot disease and was used as the wild-type strain. FP9 displays high fumonisin production and strong pathogenicity [13]. The pCPXHY2GFP plasmid encoding GFP was kindly provided by Professor Chen Baoshan from Guangxi University, Nanning, China [51], and the pBluescript II SK plasmid was purchased from Agilent Technologies. Colony morphology was compared on potato dextrose agar (PDA) medium, and radial growth was determined by measuring colony diameters after three days of growth on PDA. Fungal genomic DNA was isolated from mycelia grown in potato dextrose broth (PD), and conidia were produced for inoculum by growing the fungus in liquid YEPD medium (3 g yeast extract, 10 g peptone, and 20 g dextrose for 1 L). 

### 4.2. Gene Deletion Constructs, Transformation, and Southern Blot Analysis

In this study, the encoded *FUM1*, *FUM6*, *FUM8*, *FUM19*, and *FUM21* sequences of *F. proliferatum* (GenBank accession PRJNA517537) respectively displayed 83.00%, 85.03%, 77.90%, 80.07%, and 74.74% nucleotide identities compared with those from *F. verticillioides* (GenBank accession AF155773) [17,22]. Deletion constructs for *FUM1*, *FUM6*, *FUM8*, *FUM19*, and *FUM21* were engineered utilizing the approximately 1.5–2.0 kb upstream and downstream regions flanking the coding region. For each gene, the upstream/downstream regions and hygromycin B resistance gene (*HygR*) were amplified by PCR from genomic DNA and plasmid DNA (pCPXHY2GFP), respectively. The three fragments were then integrated into the plasmid pBluescript II SK using a multi-fragment recombination kit (Yeasen Biotech, Shanghai, China). Appendix A lists the primers used to amplify each region of each gene. The resulting vectors were transformed separately into the wild-type strain FP9 as described previously by Sun et al. [31]. Primary transformants were first screened for hygromycin resistance (200 mg/mL), then by PCR to determine whether double homologous recombination, and therefore gene deletion, had occurred. For each *FUM* gene analyzed in this study, three independently isolated deletion mutants, as determined by PCR, were selected for Southern blot analysis to confirm deletion of the targeted *FUM* gene and for LC-MS analysis to determine the effect of the *FUM* gene deletion on fumonisin production. For Southern blots, genomic DNA was isolated and digested with *EcoRV*, then the digested DNA was separated on a 0.8% agarose gel by electrophoresis and blotted onto a nylon membrane. In the engineered strains, the *FUM* gene was replaced by the *HygR* coding region, which was labeled with DIG using the DIG-High Prime Labeling and Detection Kit II and hybridized to the Southern blot [33]. Detection and visualization procedures were carried out following the manufacturer’s manual.

### 4.3. Analysis of Fumonisin Production

For each strain, the concentration of the conidia solution was adjusted to approximately 3 × 10^6^ spores/mL, and 1 mL of the conidia suspension was inoculated into RG medium (20 g rice grains and 20 mL water per bottle). In order to assess fumonisin production in the *FUM* deletion mutants, strains were cultured on sterilized RG at 28 °C for seven days. To further explore the function of the *FUM* genes, five *FUM* mutant strains (Δ*FUM1*, Δ*FUM6*, Δ*FUM8*, Δ*FUM19*, and Δ*FUM21*) were co-cultivated to assess the resulting effects on fumonisin synthesis. For example, a total of 1 mL conidia suspension was inoculated with a 1:1 mixture of Δ*FUM6* + Δ*FUM1* or Δ*FUM6* + Δ*FUM21.* All fumonisin B_1_ (FB_1_) induction experiments were performed using three biological replicates. The fumonisin content of culture extracts was determined by high performance liquid chromatography tandem mass spectrometry (HPLC-MS/MS), as described by Li et al. [35]. The sample was extracted with a methanol–water–acetic acid (74:25:1) solution and cleaned using a strong anion exchange column (SAX, Agilent Technologies, Santa Clara, CA, USA). FB_1_ was completely separated on a ZORBAX Extend-C_18_ column (150 × 2.1 mm, 1.8 μm, Agilent Technologies, Santa Clara, CA, USA) with gradient elution using a 0.1% acetic acid–water solution and acetonitrile as the mobile phases, then detected by positive-ion electrospray ionization mass spectrometry under select reaction monitoring mode. The experiment was independently repeated three times.

### 4.4. Virulence Assay

The susceptible rice (*Oryza sativa* subsp. japonica) cultivar Xiushui 134 was used for all plant pathogenicity assays. We modified the method described by Sun to develop our in vivo assay [13]. Briefly, at the pollen cell meiosis to maturing stages, three panicles of rice were inoculated with wild-type (WT) and mutant strains at a concentration of 5 × 10^5^ spores/mL. Infected rice plants were placed in a growth chamber with controlled temperature (28 °C), relative humidity (80%), and light cycling (16 h light and 8 h dark). Rice plants inoculated with sterile water served as the control. Fourteen days after inoculation, the panicle was sampled and observed for symptoms of RSRD. The RSRD index as defined by Huang was used to determine the index for panicle infection. Grades were defined as follows: grade 0, no grains infected; grade 1, 0.1–10.0% grains infected; grade 3, 10.1–25.0% grains infected; grade 5, 25.1–50.0% grains infected; grade 7, 50.1–75.0% grains infected; grade 9, >75.1% grains infected [30]. The disease index was calculated according to the following formula:
$$DI=\frac{\sum \left(Bi\times Bd\right)}{M\times Md}\times 100$$
where *DI* is the RSRD disease index, *Bi* is number of panicles with grade I, *Bd* is the individual grade value (0 to 9), *M* is total number of observed panicles, and *Md* is the maximum grade value.

### 4.5. qRT-PCR Expression Analysis 

For each strain, the conidia solution described for fumonisin production was inoculated into 50 mL PD medium and incubated in a rotary shaker (180 r/min) at 28 °C for three days. The mycelium was collected, and 1 g (wet weight) was re-inoculated into 50 mL GAPL medium (3 g KH_2_PO_4_, 0.3 g MgSO_4_, 5 g NaCl, 6 mM glutamine, and 20 g sucrose in 1 L). We then collected mycelium at different time points for RT-PCR analysis to assess *FUM* gene expression. From this data, we identified the culture time associated with high gene expression and used this time point for future experiments. Amplification efficiency (AE) data for the oligonucleotide pairs used for qPCR analysis are shown in Appendix A. The AEs of the *TUB*, *FUM1*, *FUM6*, and *FUM8* primer sets were 0.99, 0.98, 0.98, and 0.94, respectively.

Total RNA samples were isolated from vegetative hyphae from GAPL cultures sampled at 3, 8, 12, 16, 24, 36, 48, and 72 h using the RNAiso Plus reagent (Takara, Dalian, China). For qRT-PCR analysis, first-strand cDNA was synthesized using the PrimeScipt RT Reagent Kit with a gDNA Eraser (Takara, Dalian, China) following the manufacturer’s instructions. The β-tubulin gene (*TUB2*) was used as an endogenous control to normalize differences in mRNA quantity due to differing amounts of total RNA. The expression levels of each gene were calculated using the 2^−ΔΔ*Ct*^ method. Data from three biological replicates were used to calculate the mean and standard deviation. The data were processed and plotted in Microsoft Excel (version 2003, Microsoft, Redmond, WA, USA) and Graphpad software (version 6.0, GraphPad, San Diego, CA, USA). The Duncan method in SAS statistical software (version 9.2, SAS Institute, Cary, NC, USA) was used to test for significant differences.

## Figures and Tables

**Figure 1 toxins-11-00327-f001:**
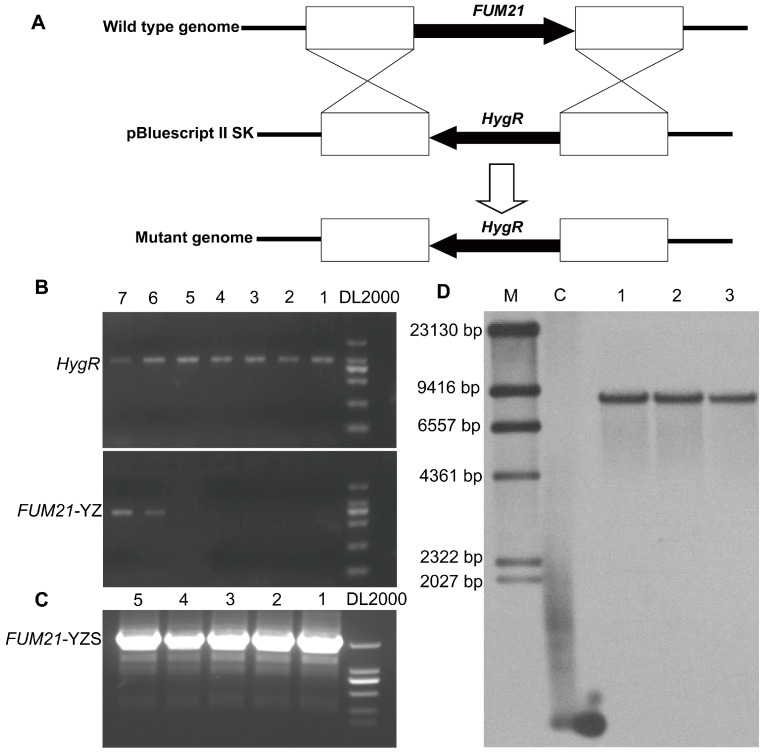
Acquisition of *Fusarium proliferatum FUM* knockout mutants. (**A**) Disruption of *FUM21*; (**B**) PCR detection of *FUM21* disruption, *HygR*-F/*HygR*-R, *FUM21*-YZ-F/*FUM21*-YZ-R are primers used for 1357-bp and 714-bp fragments of hygromycin resistance and *FUM21* gene sequence, respectively (1–5: positive transformants, 6 and 7: negative transformants); (**C**) PCR detection of *FUM21* disruption following subculture (5 generations), *FUM21*-YZS-F and *FUM21*-YZS-R are primers used for 1966-bp fragment across part of the hygromycin resistance and the homologous recombination sequence upstream of the gene amplification product (1-5: positive transformants); (**D**) Southern analysis of wild-type (WT) and knockout transformants (M: DNA molecular weight marker; C: *HygR* positive control; lanes 1, 2, and 3: *FUM21* mutants).

**Figure 2 toxins-11-00327-f002:**
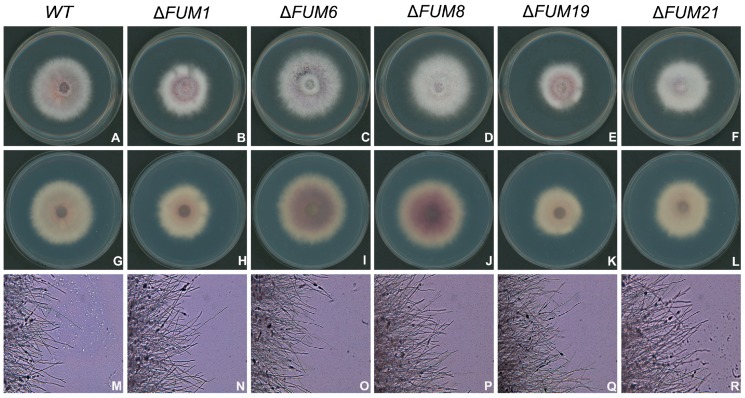
Colony morphology phenotypes of wild-type (WT) and *FUM* deletion strains grown on PDA medium for 3 days (front: **A**–**F**, back: **G**–**L**); morphology of mycelium at the colony edge after growth on PDA medium for 16 h (**M**–**R**).

**Figure 3 toxins-11-00327-f003:**
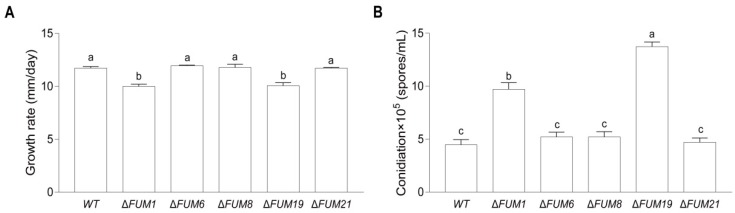
Growth rates of the wild-type (WT) and *FUM* mutant strains on PDA medium (**A**) and conidiation in liquid YEPD medium (**B**). Mean and standard deviation were calculated with data from three independent biological replicates. Different lowercase letters in the same graph indicate significant differences at the 5% level.

**Figure 4 toxins-11-00327-f004:**
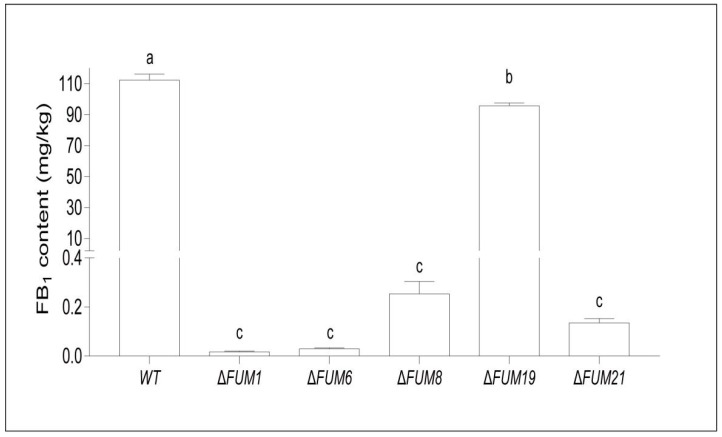
Fumonisin (FB_1_) production in wild-type (WT) and *FUM* mutant strains. Mean and standard deviation were calculated with data from three independent biological replicates. Different lowercase letters in the same graph indicate significant differences at the 5% level.

**Figure 5 toxins-11-00327-f005:**
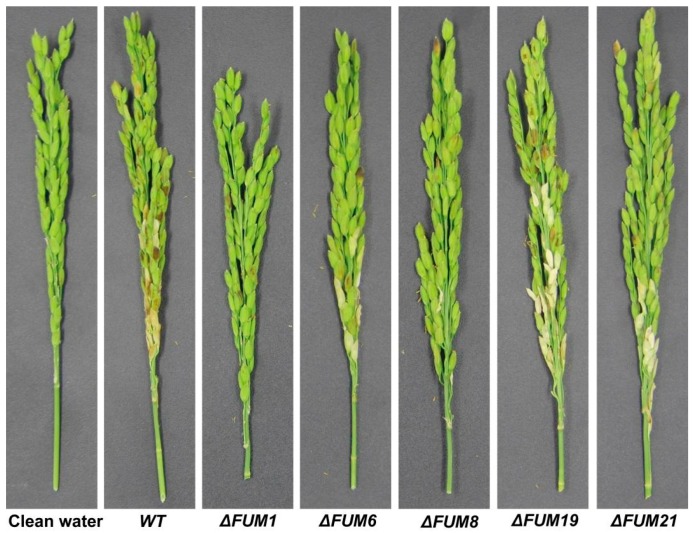
Virulence assays were performed on Xiushui134 spikelets 14 days after inoculation.

**Figure 6 toxins-11-00327-f006:**
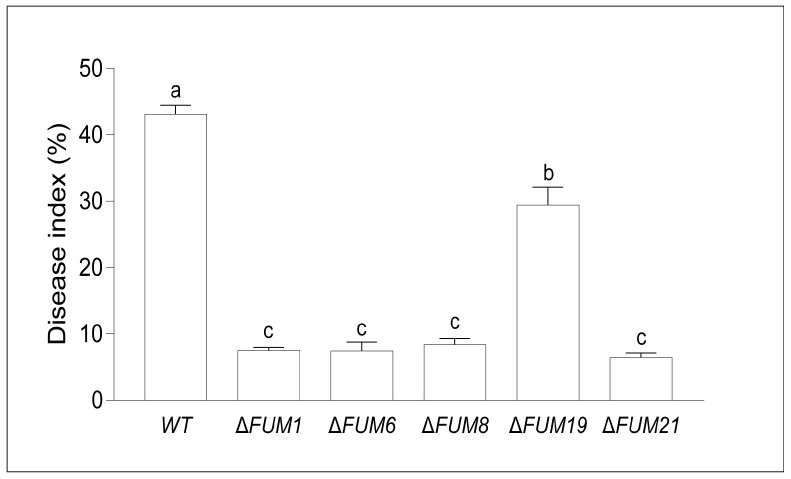
Rice spikelet rot disease indexes for wild-type and *FUM* mutant strains. Mean and standard deviation were calculated with data from three independent biological replicates. Different lowercase letters in the same graph indicate significant differences at the 5% level.

**Figure 7 toxins-11-00327-f007:**
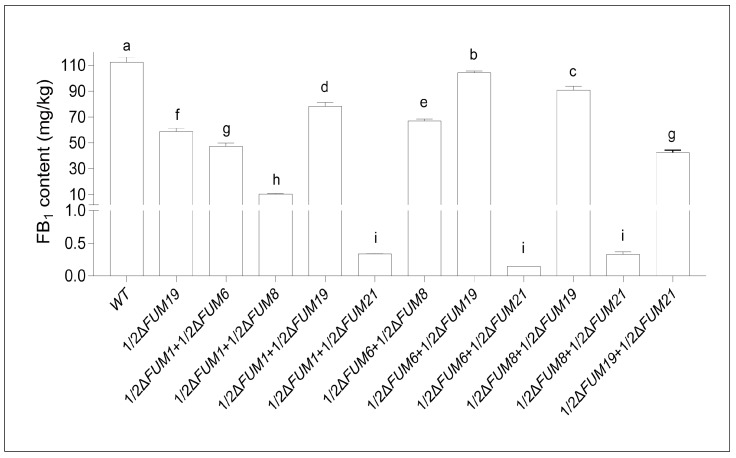
Fumonisin (FB_1_) production in mutants co-cultured on solid RG medium. Mean and standard deviation were calculated with data from three independent biological replicates. Different lowercase letters in the same graph indicate significant differences at the 5% level.

**Figure 8 toxins-11-00327-f008:**
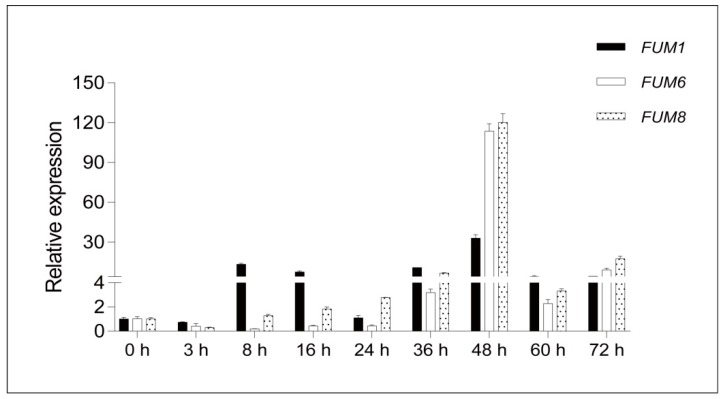
Relative expression of *FUM1*, *FUM6*, and *FUM8* at different culture times. Mean and standard deviation were calculated with data from three independent biological replicates.

**Figure 9 toxins-11-00327-f009:**
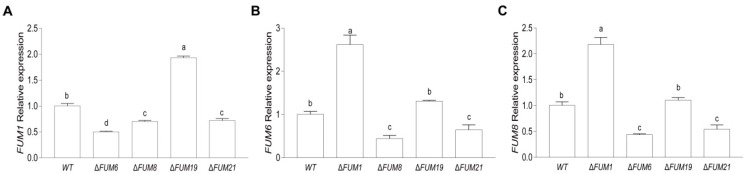
Expression of *FUM1* (**A**), *FUM6* (**B**), and *FUM8* (**C**) in each *FUM* mutant strain. Mean and standard deviation were calculated with data from three independent biological replicates. Different lowercase letters in the same graph indicate significant differences at the 5% level.

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
