# Peer review of "Effects of Disruption of Five FUM Genes on Fumonisin Biosynthesis and Pathogenicity in Fusarium proliferatum"

_toxins, 2019, doi:10.3390/toxins11060327_

Round 1

Reviewer 1 Report

Article entitled “Effects of five FUM genes on fumonisin biosynthesis and pathogenicity in Fusarium proliferatum”, describes the isolation of several FUM genes deletion mutants in F. proliferatum.

Authors analyze in these mutants their ability to produce fumonisins, and try to correlate this parameter with the induction of RSRD in rice plants.

Even, when the article includes a considerable amount or work, in general well performed, the findings of it are mostly expected and known based on similar results obtained with close Fusarium species.

Also some procedures have to be improved (e.g. qPCR), or more detailed (e.g. data about multifragment recombination kit). Some data are missing (e.g. experimental data about the isolation of at least three of the deleted mutants). No mention to any supplementary material is included.

The justification for the selection of these particular FUM genes is missing. 

Abstract

Line 16.- do not use the acronym for RSRD, first time cited authors must include the extended name.

Line 58.- Fig. 1 does not correspond with the text included in this part of the text.

Line 64.- Authors have to justify why they selected FUM1, 6, 8, 9 and 21 genes and no others?

Materials and Methods

Line 93.- Authors have to include what kit was used for multi-fragment recombination experiments.

Section 2.5.- Have authors calculated the amplification efficiency of the oligonucleotide pairs used for qPCR analysis?. These data must be included in the article.

Results

Line 155.- authors cite Fig. 2, but I guess is Fig. 1. Please check.

Section 3.1. Only data corresponding to the isolation of FUM21 deleted mutant were included. Are the data for the isolation of the other mutants included as supplementary material?, I strongly recommend to do i

Fig. 1. I keep some concern about that the PCR and Southern analyses included in the article should be enough to be sure that the mutants are real deleted mutants. At least a PCR with the borders of the deletion cassette once integrated in the genome, and the sequence of the PCR fragments would be required.

Fig. 2 and Fig. 3.- Authors have to think if these two Figures are really required for the main file. I propose to transfer them to the suplementary material, and keep in the main text the description as it is in the current version.

Lines 189-190.- Regarding produciton of fumonisin in the fum19 mutant. Have authors thought to quantify intracellular fumonisins in this mutant?

Discussion

Paragraph from lines 327 to 341.- I do not see the need to include so much data related to trichothecenes production by Fusarium in this article. Please try to find a correlation of these data with the present article, or delete these sentences.

The concluding paragraph seems to be a statement about future experiments more than a conclusion of the current work. Please modify to state the main conclusions (usualy few) extracted from this article.

Author Response

Reviewer #1

Question 1: ABSTRACT: Line 16.- do not use the acronym for RSRD, first time cited authors must include the extended name.

Answer: Thank you

Reviewer #1

Question 1: ABSTRACT: Line 16.- do not use the acronym for RSRD, first time cited authors must include the extended name.

Answer: Thank you for your careful review. We added an explanation of RSRD to the abstract, at line 17: “rice spikelet rot disease (RSRD)”.

Question 2: INTRODUCTION: Line 58.- Fig. 1 does not correspond with the text included in this part of the text.

Answer: Thank you for your careful review. The error has been deleted.

Question 3: INTRODUCTION: Line 64.- Authors have to justify why they selected FUM1, 6, 8, 9 and 21 genes and no others?

Answer: Thank you for your careful review. We have included additional text to explain our choice of genes: (lines 51-55): “FUM21 is a regulator of fumonisin synthesis. FUM1, FUM6, and FUM8 are key genes in the synthesis of fumonisin precursors and FUM19 functions in fumonisin transport. In this study, we chose to investigate these five genes to broadly understand the function of fumonisin synthesis genes and to observe the effects of disruption of these genes on various aspects of fumonisin synthesis.”

Question 4: MATERIALS AND METHODS: Line 93.- Authors have to include what kit was used for multi-fragment recombination experiments.

Answer: Thank you for your careful review. The product and vendor information (Yeasen Biotech, China) has been added to the manuscript at line 329.

Question 5: MATERIALS AND METHODS: Section 2.5.- Have authors calculated the amplification efficiency of the oligonucleotide pairs used for qPCR analysis?. These data must be included in the article.

Answer: Thank you for your helpful suggestion. The following text has been added to the manuscript (lines 384-386): “Amplification efficiency (AE) data for the oligonucleotide pairs used for qPCR analysis are shown in Figure S1. The AEs of the TUB, FUM1, FUM6, and FUM8 primer sets were 0.99, 0.98, 0.98, and 0.94, respectively.”

Question 6: RESULTS: Line 155.- authors cite Fig. 2, but I guess is Fig. 1. Please check.

Answer: Thank you for your careful review. The error has been corrected at line 89.

Question 7: RESULTS: Only data corresponding to the isolation of FUM21 deleted mutant were included. Are the data for the isolation of the other mutants included as supplementary material?, I strongly recommend to do 

Answer: Thank you for your suggestion. Because the sizes of the resulting PCR products and Southern blot bands are almost the same between all the mutants, there are no obvious differences to show in the figure. The specific sizes of the other products can be found in the primer list in the Supplementary Materials.

Question 8: RESULTS: Fig. 1. I keep some concern about that the PCR and Southern analyses included in the article should be enough to be sure that the mutants are real deleted mutants. At least a PCR with the borders of the deletion cassette once integrated in the genome, and the sequence of the PCR fragments would be required.

Answer: Thank you for your helpful suggestion. We have indeed performed the PCR experiment you describe. The first round of PCR verification included 2 repetitions, and then we further tested the strain after subculture. Because all of the results from these experiments were consistent, we did not originally include them in the manuscript. However, we have now added text describing these results at lines 94-98: “After sub-culturing for five generations, we used a pair of new primers (FUM21-YZS) to PCR across part of the hygromycin resistance gene and the homologous recombination sequence upstream of the gene to further verify loss of the gene of interest. Using this approach, we found that the 5 mutants stably inherited their respective gene deletions (Fig 1 C).” 

Question 9: RESULTS: Fig. 2 and Fig. 3.- Authors have to think if these two Figures are really required for the main file. I propose to transfer them to the suplementary material, and keep in the main text the description as it is in the current version.

Answer: Thank you very much for your suggestion. As we believe these two figures present important data for the manuscript’s conclusions, we hope that they will maintain their placement in the main text. Per the suggestion of another reviewer, the images in these figures have been modified.

Question 10: RESULTS: Lines 189-190.- Regarding produciton of fumonisin in the fum19 mutant. Have authors thought to quantify intracellular fumonisins in this mutant?

Answer: Thank you for your helpful suggestion. This quantitative data has been added to the manuscript at lines 130-132: “The ΔFUM19 strain was able to synthesize 95.43 mg/kg of fumonisin, which represented only a 14.7% reduction in fumonisin synthesis when compared to the wild-type (111.83 mg/kg).”

Question 11: DISCUSSION: Paragraph from lines 327 to 341.- I do not see the need to include so much data related to trichothecenes production by Fusarium in this article. Please try to find a correlation of these data with the present article, or delete these sentences.

Answer: Thank you for your careful review and helpful suggestion. Because DON is an important toxin in Fusarium, and the DON-related research is more in-depth than that for fumonisin, we reference DON research methods in our fumonisin studies. We have modified the text to explain and enhance the relevance of DON/trichothecene research to our findings (lines 208-210 and 284-287): “Similar results were found in another Fusarium toxin study that identified TRI6 and TRI10 as transcription factors in the trichothecenes biosynthetic pathway” and “The Fusarium graminearum trichothecene precursor synthase gene TRI5 is the first and most well-studied pathogenicity factor and is a key enzyme in the first step of the synthesis of deoxynivalenol (DON), another important secondary metabolite.” 

Question 12: DISCUSSION: The concluding paragraph seems to be a statement about future experiments more than a conclusion of the current work. Please modify to state the main conclusions (usualy few) extracted from this article.

Answer: Thank you for your helpful suggestion. The final paragraph of the discussion section has been completely rewritten to function as a summary of our work (lines 299-308): “In our study, we found that FUM1, FUM6, FUM8, and FUM21 are essential for fumonisin synthesis in F. proliferatum, while FUM19 is non-essential. Loss of fumonisin synthesis is associated with a decrease in pathogenicity. Partial mutant co-culture can restore fumonisin synthesis. Whether this restoration is due to exchange of fumonisin intermediates and how these intermediates are utilized by each strain requires further research. Our comprehensive analysis suggests that the functions of these five FUM genes in F. proliferatum are consistent with those observed in F. verticillioides. In future research, we will utilize overexpression of FUM proteins and assessment of additional FUM genes to further describe FUM gene functions and the fumonisin biosynthetic pathway. This research will support efforts to reduce fumonisin contamination in food production.”

for your careful review. We added an explanation of RSRD to the abstract, at line 17: “rice spikelet rot disease (RSRD)”.

Question 2: INTRODUCTION: Line 58.- Fig. 1 does not correspond with the text included in this part of the text.

Answer: Thank you for your careful review. The error has been deleted.

Question 3: INTRODUCTION: Line 64.- Authors have to justify why they selected FUM1, 6, 8, 9 and 21 genes and no others?

Answer: Thank you for your careful review. We have included additional text to explain our choice of genes: (lines 51-55): “FUM21 is a regulator of fumonisin synthesis. FUM1, FUM6, and FUM8 are key genes in the synthesis of fumonisin precursors and FUM19 functions in fumonisin transport. In this study, we chose to investigate these five genes to broadly understand the function of fumonisin synthesis genes and to observe the effects of disruption of these genes on various aspects of fumonisin synthesis.”

Question 4: MATERIALS AND METHODS: Line 93.- Authors have to include what kit was used for multi-fragment recombination experiments.

Answer: Thank you for your careful review. The product and vendor information (Yeasen Biotech, China) has been added to the manuscript at line 329.

Question 5: MATERIALS AND METHODS: Section 2.5.- Have authors calculated the amplification efficiency of the oligonucleotide pairs used for qPCR analysis?. These data must be included in the article.

Answer: Thank you for your helpful suggestion. The following text has been added to the manuscript (lines 384-386): “Amplification efficiency (AE) data for the oligonucleotide pairs used for qPCR analysis are shown in Figure S1. The AEs of the TUB, FUM1, FUM6, and FUM8 primer sets were 0.99, 0.98, 0.98, and 0.94, respectively.”

Question 6: RESULTS: Line 155.- authors cite Fig. 2, but I guess is Fig. 1. Please check.

Answer: Thank you for your careful review. The error has been corrected at line 89.

Question 7: RESULTS: Only data corresponding to the isolation of FUM21 deleted mutant were included. Are the data for the isolation of the other mutants included as supplementary material?, I strongly recommend to do 

Answer: Thank you for your suggestion. Because the sizes of the resulting PCR products and Southern blot bands are almost the same between all the mutants, there are no obvious differences to show in the figure. The specific sizes of the other products can be found in the primer list in the Supplementary Materials.

Question 8: RESULTS: Fig. 1. I keep some concern about that the PCR and Southern analyses included in the article should be enough to be sure that the mutants are real deleted mutants. At least a PCR with the borders of the deletion cassette once integrated in the genome, and the sequence of the PCR fragments would be required.

Answer: Thank you for your helpful suggestion. We have indeed performed the PCR experiment you describe. The first round of PCR verification included 2 repetitions, and then we further tested the strain after subculture. Because all of the results from these experiments were consistent, we did not originally include them in the manuscript. However, we have now added text describing these results at lines 94-98: “After sub-culturing for five generations, we used a pair of new primers (FUM21-YZS) to PCR across part of the hygromycin resistance gene and the homologous recombination sequence upstream of the gene to further verify loss of the gene of interest. Using this approach, we found that the 5 mutants stably inherited their respective gene deletions (Fig 1 C).” 

Question 9: RESULTS: Fig. 2 and Fig. 3.- Authors have to think if these two Figures are really required for the main file. I propose to transfer them to the suplementary material, and keep in the main text the description as it is in the current version.

Answer: Thank you very much for your suggestion. As we believe these two figures present important data for the manuscript’s conclusions, we hope that they will maintain their placement in the main text. Per the suggestion of another reviewer, the images in these figures have been modified.

Question 10: RESULTS: Lines 189-190.- Regarding produciton of fumonisin in the fum19 mutant. Have authors thought to quantify intracellular fumonisins in this mutant?

Answer: Thank you for your helpful suggestion. This quantitative data has been added to the manuscript at lines 130-132: “The ΔFUM19 strain was able to synthesize 95.43 mg/kg of fumonisin, which represented only a 14.7% reduction in fumonisin synthesis when compared to the wild-type (111.83 mg/kg).”

Question 11: DISCUSSION: Paragraph from lines 327 to 341.- I do not see the need to include so much data related to trichothecenes production by Fusarium in this article. Please try to find a correlation of these data with the present article, or delete these sentences.

Answer: Thank you for your careful review and helpful suggestion. Because DON is an important toxin in Fusarium, and the DON-related research is more in-depth than that for fumonisin, we reference DON research methods in our fumonisin studies. We have modified the text to explain and enhance the relevance of DON/trichothecene research to our findings (lines 208-210 and 284-287): “Similar results were found in another Fusarium toxin study that identified TRI6 and TRI10 as transcription factors in the trichothecenes biosynthetic pathway” and “The Fusarium graminearum trichothecene precursor synthase gene TRI5 is the first and most well-studied pathogenicity factor and is a key enzyme in the first step of the synthesis of deoxynivalenol (DON), another important secondary metabolite.” 

Question 12: DISCUSSION: The concluding paragraph seems to be a statement about future experiments more than a conclusion of the current work. Please modify to state the main conclusions (usualy few) extracted from this article.

Answer: Thank you for your helpful suggestion. The final paragraph of the discussion section has been completely rewritten to function as a summary of our work (lines 299-308): “In our study, we found that FUM1, FUM6, FUM8, and FUM21 are essential for fumonisin synthesis in F. proliferatum, while FUM19 is non-essential. Loss of fumonisin synthesis is associated with a decrease in pathogenicity. Partial mutant co-culture can restore fumonisin synthesis. Whether this restoration is due to exchange of fumonisin intermediates and how these intermediates are utilized by each strain requires further research. Our comprehensive analysis suggests that the functions of these five FUM genes in F. proliferatum are consistent with those observed in F. verticillioides. In future research, we will utilize overexpression of FUM proteins and assessment of additional FUM genes to further describe FUM gene functions and the fumonisin biosynthetic pathway. This research will support efforts to reduce fumonisin contamination in food production.”

Reviewer 2 Report

Dear Authors,

the paper describes the effect of FUM gene deletion on fumonisin biosynthesis and pathogenicity in F. proliferatum. The topic is quite interesting since FUM gene deletion has never been studied in this species. Results could contribute to widen the knowledge on genetic determinants and fumonisin production in this species. The experimental methods are quite suitable, but I think some experiment should be performed again with some modifications. The authors should avoid the use of term speculation since it in not possible that every conclusion made is based on a speculation. Most importantly the writing surely needs a revision; in the results section, too many times there are included comments or speculation and the discussion must be completely rewritten, several statements are poorly related to the results, other are nonsense, other speculations.

TITLE

I suggest modifying it, since the author did not evaluate the effect of FUM gene but the effect of their deletion.

ABSTRACT

Line 16 Please explain RSRD

Line 16-18 Please rewrite this statement, too forced. You do not have any evidence to state that there is an exchange of enzymes or intermediates between strains.

MATERIALS AND METHODS

The experimental procedure should be implemented, and also additional missing data must be included in the manuscript.

2.3 What about other fumonisin analogues? Why they have not been included in the analysis?

2.3 Usually FB production is measured after a longer incubation (14 days) especially if performed not in an artificial medium but on grains.

2.3 In the co-cultivation experiments, since the authors did not perform any growth experiment on rice but only on PDA, the authors should include a quantitative analysis (real time or qPCR) to evaluate the growth rate of each mutant strains. Otherwise it is not possible to conclude that the effect is due to the co-cultivation.  

2.4 How many plants have been inoculated for each strain?

2.5 A table must be included describing the primers used and the size of the amplicons obtained.

The paragraph related to the statistical analysis is missing

RESULTS

3.2 Do the authors have any indication of the stability of the deletion?

3.2 Fig. 1 S-X is not clear. It can be deleted.

3.3 Statements about the relevance of the results or conclusions must be moved in the appropriate section

3.4 From Fig. 5 is seems that also ΔFum21 and ΔFum6 has some pathogenicity potential, indeed the disease index is not so low. Authors must carefully report and explain their results.

3.5 Authors cannot speculate anything in the result section (lines 219 and 223)

3.6 Where are the results of FUM21 gene relative expression? Why they have not been performed/included?

DISCUSSION

Overall it in not incisive and it is not clear which is the innovation brought by this paper nor the importance of the results showed in the global context of fumonisin contamination and the pathogenicity of Fusarium proliferatum towards rice.

Lines 259-261. Authors cannot speculate (again) this since they did not incubate the mutants in any nutrient-limiting condition or stressing environment.

Lines 270-273. Not true. The fact that when the ΔFum21 mutant was co-culterd with the other mutants there is no significant change in FB biosynthesis does not imply that Fum21 is the regulator,  even if this can be deduced by its homology in other species or from the fact that it is a Zn(II)Cys6 binding protein.  

Line 274. The sentence is obvious but not true, indeed the deletion of Fum21 does not completely block the expression of gene.

Line 289. Please describe the putative function of FUM12 and FUM15. “complement” is not appropriate. Do the authors mean compensate? It is not a complementation.

Lines 291-294. The authors does not provide any evidence of a possible exchange of fumonisin intermediate or enzyme exchange between strains.

Lines 296-298 This sentence has no sense

Line 306 Authors must specify in with species the deletion of FUM6 and FUM8 blocks the synthesis of fumonisin intermediates and a reference is needed

Lines 320-323. This statement has no sense. Why if FB is accumulated and create a “stressful environment” should F. proliferatum be induced to produce more FB?

Author Response

Reviewer #2  

Question 1: TITLE: I suggest modifying it, since the author did not evaluate the effect of FUM gene but the effect of their deletion.

Answer: We appreciate your useful suggestion. We have modified the title of the paper to “Effects of disruption of five FUM genes on fumonisin biosynthesis and pathogenicity in Fusarium proliferatum”.

Question 2: ABSTRACT: Line 16 Please explain RSRD.

Answer: Thank you for your careful review. We added an explanation of RSRD to the abstract (“rice spikelet rot disease (RSRD)”) at line 17.

Question 3: ABSTRACT: Line 16-18 Please rewrite this statement, too forced. You do not have any evidence to state that there is an exchange of enzymes or intermediates between strains.

Answer: Thank you for your suggestion. We rephrased this statement as, “Taken together, our results demonstrate that FUM1, FUM6, FUM8, and FUM21 are essential for fumonisin synthesis, and FUM19 is non-essential. Partial mutants lost the ability to synthesize fumonisin, and co-culture of the mutants was able to restore fumonisin biosynthesis” (lines 17-20).

Question 4: MATERIALS AND METHODS: 2.3 What about other fumonisin analogues? Why they have not been included in the analysis? 

Answer: We appreciate your thoughtful suggestion. In previous research performed by us and others, FB1 was found to be more abundant than FB2 and FB3 in F. proliferatum. Thus, we chose to measure only FB1 in subsequent experiments. We added a comment regarding this choice to the introduction (“Therefore, in our study, we chose to use FB1 as a representative of total fumonisin content”) at lines 38-39.

Question 5: MATERIALS AND METHODS: 2.3 Usually FB production is measured after a longer incubation (14 days) especially if performed not in an artificial medium but on grains?

Answer: Thank you for your useful suggestion. In previous studies, we observed high fumonisin content in rice grain medium after 7-day incubation. We also observed higher fumonisin content in the rice medium than in the corn medium. For these reasons, we chose to measure after 7 days of culture in rice grain medium.

Question 6: MATERIALS AND METHODS: 2.3 In the co-cultivation experiments, since the authors did not perform any growth experiment on rice but only on PDA, the authors should include a quantitative analysis (real time or qPCR) to evaluate the growth rate of each mutant strains. Otherwise it is not possible to conclude that the effect is due to the co-cultivation.

Answer: We appreciate your useful suggestion. To our knowledge, there are currently no genes in our strain that can be used to measure biomass. However, in another experiment, we measured ergosterol production and found the levels to be almost indistinguishable between strains. Thus, we are confident that our experimental method is reliable.

Question 7: MATERIALS AND METHODS: 2.4 How many plants have been inoculated for each strain? 

Answer: We appreciate your careful review. Three rice panicles were inoculated, as now indicated at line 362.

Question 8: MATERIALS AND METHODS: 2.5 A table must be included describing the primers used and the size of the amplicons obtained.

Answer: Thank you for your suggestion. We have added a table with this information to the Supplementary Materials. Table S1 lists the primers used to amplify each region of each gene, as indicated at lines 329-330.

Question 9: MATERIALS AND METHODS: The paragraph related to the statistical analysis is missing.

Answer: We thank you for your careful review and apologize for the omission. We have added language describing our analyses at lines 393-395: “The data were processed and plotted in Microsoft Excel 2003 and Graphpad6 software. The Duncan method in SAS 9.2 statistical software was used to test for significant differences.”

Question 10: RESULTS: 3.2 Do the authors have any indication of the stability of the deletion?

Answer: We appreciate your question. To address the stability of the gene deletions, performed subculture experiments. Following subculture, the genetic deletion and detected phenotype were stable. We have added the following language to the manuscript, at lines 94-98: “After sub-culturing for five generations, we used a pair of new primers (FUM21-YZS) to PCR across part of the hygromycin resistance gene and the homologous recombination sequence upstream of the gene to further verify loss of the gene of interest. Using this approach, we found that the 5 mutants stably inherited their respective gene deletions (Fig 1 C).”

Question 11: RESULTS: 3.2 Fig. 1 S-X is not clear. It can be deleted.

Answer: Thank you for your suggestion. Fig. 1 S-X has been deleted.

Question 12: RESULTS: 3.3 Statements about the relevance of the results or conclusions must be moved in the appropriate section.

Answer: Per your suggestion, we have moved these conclusions to the discussion section (lines 299-300).

Question 13: RESULTS: From Fig. 5 is seems that also ΔFum21 and ΔFum6 has some pathogenicity potential, indeed the disease index is not so low. Authors must carefully report and explain their results. 

Answer: We appreciate your careful review. Because the rice was grown outdoors prior to inoculation, the white spikelet may be caused by physiological effects, such as high temperature. Upon further investigation of this phenotype in our mutants, we observed an area of red lesions on the grain. Thus, the original low disease index was due to an error in our assessment and has been revised (lines 144-148).

Question 14: RESULTS: 3.5 Authors cannot speculate anything in the result section (lines 219 and 223)

Answer: Thank you for your careful review. We have added new data and the relevant text has been modified (lines 156-160): “However, ΔFUM21 co-cultured with ΔFUM19 (1:1 ratio) was able to produce fumonisin; this co-culture produced 72.3% of the fumonisin produced by ΔFUM19 cultured alone. In constrast, three other co-cultures (ΔFUM1+ΔFUM19, ΔFUM6+ΔFUM19, and ΔFUM8+ΔFUM19; all in 1:1 ratios) produced 133.0%, 177.4%, and 155.4% of the fumonisin produced by ΔFUM19 cultured alone, greater than that observed for the ΔFUM19+ΔFUM21 co-culture”.

Question 15: RESULTS: Where are the results of FUM21 gene relative expression? Why they have not been performed/included?

Answer: Thank you for your helpful suggestion. Because only the FUM1-, FUM6-, and FUM8-deletion strains were co-cultured to assess restoration of fumonisin synthesis, we limited our expression analyses assess to these three genes. However, we will address FUM21 expression in future experiments. We have explained the reason for limiting our expression analyses to FUM1, FUM6, and FUM8 at lines 168-171: “From our results, we can conclude that FUM1, FUM6, and FUM8 are important for fumonisin synthesis, and that these genes’ effects are somewhat interdependent. Therefore, we selected to quantitatively assess the expression of these three genes to test if the observed changes in fumonisin synthesis in these mutants are due to regulatory changes in gene expression.”

Question 16: DISCUSSION: Lines 259-261. Authors cannot speculate (again) this since they did not incubate the mutants in any nutrient-limiting condition or stressing environment.

Answer: Thank you for your suggestion. The cited text has been revised (lines 200-204): “Deletion of the evaluated FUM genes has no effect on the morphology of the mycelium at the edge of the colony, but deletion of FUM1 and FUM19 influences growth and conidiation, likely via changes in metabolism or certain growth-related genes. Therefore, given the increase in sporulation observed for the FUM mutants, we speculate that FUM1 and FUM19 may negatively regulate sporulation.”

Question 17: DISCUSSION: Lines 270-273. Not true. The fact that when the ΔFum21 mutant was co-culterd with the other mutants there is no significant change in FB biosynthesis does not imply that Fum21 is the regulator, even if this can be deduced by its homology in other species or from the fact that it is a Zn(II)Cys6 binding protein.  

Answer: We appreciate your insightful comment. Indeed, the research presented cannot support this conclusion. I have modified the text as follows (lines 214-222): “When ΔFUM21 was co-cultured with ΔFUM19 (1:1 ratio), fumonisin synthesis was significantly reduced when compared to that of ΔFUM19 alone. However, when the other mutants were co-cultured with ΔFUM19, fumonisin synthesis was significantly increased. Therefore, fumonisin synthesis in the ΔFUM21 strain cannot be restored by co-culture with ΔFUM19. In summary, when the ΔFUM21 mutant was co-cultured with the other four mutants, we observed no significant changes in fumonisin synthesis, demonstrating that FUM21 is necessary for, but it is not directly involved in fumonisin synthesis. Deletion of FUM21 affects the expression of genes directly involved in fumonisin synthesis. Therefore, we speculate that FUM21 may play a role in regulating genes directly involved in fumonisin synthesis.”

Question 18: DISCUSSION: Line 274. The sentence is obvious but not true, indeed the deletion of Fum21 does not completely block the expression of gene.

Answer: We appreciate your careful review and helpful suggestion. We have modified the text as follows (lines 220-221): “Deletion of FUM21 can affect the expression of genes directly involved in fumonisin synthesis.”

Question 19: DISCUSSION: Line 289. Please describe the putative function of FUM12 and FUM15. “complement” is not appropriate. Do the authors mean compensate? It is not a complementation.

Answer: Thank you for your suggestion. We indeed meant “compensate,” not “complement.” The relevant text has been revised (lines 236-239): “FUM8 is the only fumonisin synthesis gene that functions as a transaminase, and FUM12 and FUM15 are able to partially compensate for the loss of FUM6 function, because FUM6p, FUM12p, and FUM15p putatively function as cytochrome P450 monooxygenases.”

Question 20: DISCUSSION: Lines 291-294. The authors does not provide any evidence of a possible exchange of fumonisin intermediate or enzyme exchange between strains.

Answer: Thank you for your careful review. As there are few studies related to enzyme or intermediate exchange in fumonisin synthesis, the conclusion mentioned is speculative and requires further research. However, there is some research to strengthen and support this hypothesis. I have modified the text as follows (lines 241-249): “We speculate that the restoration of fumonisin synthesis in co-cultures of FUM mutants is due to exchange of various enzymes or intermediates required for fumonisin synthesis between the different mutants, which compensates for the loss of function in each mutant. However, understanding the mechanisms underlying this possible exchange between mutants will require further investigation. Previous studies identified the early fumonisin biosynthetic intermediates from fungal strains with an inactivated FUM6 gene were purified using mass spectrometry and NMR spectroscopy [27]. In precursor feeding experiments, the fed intermediate was transformed into fumonisins by a F. verticillioides strain with an inactive fumonisin polyketide synthase gene, which supports our hypothesis [27].”

Question 21: DISCUSSION: Lines 296-298 This sentence has no sense

Answer: Thank you for your careful review. I have modified the text to improve clarity (lines 249-254): “When the ΔFUM1 mutant was cultured in liquid GAPL medium, expression levels of FUM6 and FUM8 were up-regulated. Based on the above findings, we conclude that FUM21 exerts a positive regulatory effect on FUM1, FUM6, and FUM8, such that when FUM1 is deleted, the regulation of FUM6 and FUM8 by FUM21p is enhanced. Moreover, GAPL medium is able to induce fumonisin synthesis and thereby induce expression of the FUM21 gene, resulting in up-regulation of FUM6 and FUM8.”

Question 22: DISCUSSION: Line 306 Authors must specify in with species the deletion of FUM6 and FUM8 blocks the synthesis of fumonisin intermediates and a reference is needed

Answer: Thank you for your suggestion. This sentence has been modified (lines 262-263): “Deletion of FUM6 or FUM8 inhibits the normal pathway of fumonisin biosynthesis in F. proliferatum.”

Question 23: DISCUSSION: Lines 320-323. This statement has no sense. Why if FB is accumulated and create a “stressful environment” should F. proliferatum be induced to produce more FB?

Answer: Thank you for your careful review and question. As there are too few related studies to support this speculation, we have modified the relevant text (lines 276-278): “A possible explanation for this result is that FUM19 exerts a negative regulatory effect on FUM1, such that when FUM19 is deleted, FUM1 expression cannot be suppressed, resulting in significant up-regulation.”

Round 2

Reviewer 1 Report

Authors have correctly addressed my initial concerns about the article.

Author Response

Thank you for your helpful suggestion.

Reviewer 2 Report

Dear authors,

I consider the revisions made suitable and the revised manuscript resulted quite improved.

I appreciate the changes made, also in the title, and I believe that further experiments will clarify to a greater extent the role of these genes in F. proliferatum.

Author Response

Thank you for your helpful suggestion.